# Evaluation of CH$_4$ Emission in Two Paddy Field Areas, Khonkaen and Ayutthaya, in Thailand

**Pongsathorn Sukdanont [1],*** , **Noppol Arunrat [2]**, **Suphachai Amkha [3]** and **Ryusuke Hatano [1]**

[1] Laboratory of Soil Science, Graduate School of Agriculture, Hokkaido University, Sapporo 060-8589, Hokkaido, Japan; hatano@chem.agr.hokudai.ac.jp

[2] Faculty of Environment and Resources Studies, Mahidol University, Nakhon Pathom 73170, Thailand; noppol.aru@mahidol.ac.th

[3] Department of Soil Science, Faculty of Agriculture at Kamphaengsaen, Kamphaengsaen Campus, Kasetsart University, Nakhon Pathom 73140, Thailand; agrscak@ku.ac.th

* Correspondence: sandsukdanont@gmail.com; Tel.: +81-(66)62-710-2211

**Simple Summary:** In anthropogenic activities, flooded paddy field is one of those events that emits CH$_4$ and that comes the necessity of leveling down CH$_4$ production from soil so the study on mechanisms of CH$_4$ production in paddy soil is vital. This study was conducted for an aim of investigating CH$_4$ influential factors in wide range of soil texture in paddy soil in Thailand and interrogating a model for predicted CH$_4$ production potential map. Our result showed that soil carbon and sandy soil are the important factors generating CH$_4$ production. To be exact, soil carbon works as a provider that implements an energy to soil microorganism to anoxically produce CH$_4$ and sandy soil, that carries less iron oxide, accelerates the CH$_4$ production. In conclusion, CH$_4$ production in paddy field under flooding period is triggered when large amount of soil carbon exists, and the reduction being processed quickly due to a smaller number of iron oxide in the soil, especially sandy soil. The knowledge of factors influencing CH$_4$ production brings about a better crop and water management. Further than that, predicted map of CH$_4$ production potential can be utilized to identify whereabout of highly emitted CH$_4$ production potential, which attention should be paid.

**Abstract:** It is well known that submerged soils emit high levels of methane (CH$_4$) due to oxygen deprivation and free iron oxide causing a quick reduction. However, there are other soil properties that control the reduction processes in soil, especially the amount of soil organic carbon (SOC). This study aimed to investigate the major factors controlling CH$_4$ production potential (CH$_4$PP) in Thai paddy fields. Two provinces, Ayutthaya, a clay soil region, and Khonkaen, a sandy soil region, were selected to represent a wide range of soil textures. Soil characteristic analysis pre- and post-incubation, and weekly gas detection in an incubation experiment over two months, was conducted. Stepwise multiple regression analysis was employed to analyze major soil factors controlling CH$_4$PP. For the regional prediction of CH$_4$PP, a map dataset of Ayutthaya and Khonkaen by the Land Development Department, Thailand, and a soil texture map (with intersected point data using the soil property map in ArcGIS) by OpenLandMap, were used. CH$_4$PP was correlated with 1:10 pH, Fe$^{2+}$, and water-soluble organic carbon (WSOC) measured after incubation. Although CH$_4$PP showed no significant correlation with any soil properties measured before incubation, CH$_4$PP was correlated with SOC, 1:10 electrical conductivity (EC), exchangeable ammonium (ExNH$_4$), and sand content. It was thought that SOC and ExNH$_4$ were related to organic matter decomposition, 1:10 EC was related to SO$_4^{2-}$ reduction and sand content was related to free oxides. Predicted regional CH$_4$PP was similar in Ayutthaya and Khonkaen, although SOC, ExNH$_4$ and 1:10 EC was higher, and sand content was lower in Ayutthaya than in Khonkaen. In both regions, the distribution of CH$_4$PP corresponded to SOC, and CH$_4$PP was lower with lower sand content and higher 1:10 EC. In clayey Ayutthaya, higher CH$_4$PP was observed in the area with higher ExNH$_4$. This indicates that soil properties other than soil texture and SOC influence CH$_4$PP in the paddy fields in Thailand.

**Keywords:** $CH_4$ production potential; incubation study; paddy field; mollisols; ultisols

## 1. Introduction

In recent decades, causes and mitigation issues related to global warming have become controversial. Methane ($CH_4$) is a stronger greenhouse gas (GHG) than $CO_2$ because it has a higher radiative trapping ability [1]. Submerged soils emit high levels of $CH_4$ due to free iron oxides causing rapid reduction reactions. In agriculture, paddy fields are an important source of atmospheric $CH_4$, as flooded conditions are preferable for proper rice growth [2,3]. Globally, rice cropping is considered to account for 5 to 20% of total $CH_4$ emission from anthropogenic actions [4]. Paddy fields in Thailand cover almost half the total agricultural land and are mostly located in the central and northeastern parts of the country. These regions have different soil types, with clayey soil in the central region, and sandy soil in the northeastern part [5,6]. Rice cultivation in Thailand was ranked as the fourth highest global $CH_4$ emitter, contributing an average of 1756.6 Gg $CH_4$ between 2010 and 2017 [7].

Several factors controlling $CH_4$ emissions have been studied, particularly in relation to plant species, cultivation practices, climate, and soil properties [8]. $CH_4$ production in the anaerobic environment of water-saturated soil depends on a variety of soil properties, both chemical and physical [8]. Under anaerobic soil conditions, methanogens process $CH_4$ through the reduction of soil oxides [2,3]. Following the depletion of oxygen, when soil microbes initiate anaerobic respiration, the order of electron acceptors use is $NO_3^-$, $MnO_2$, $Fe_2O_3$, $SO_4^{2-}$, and $CO_2$, while organic matter acts as the electron donor [9]. Inubushi et al. reported that hexose (an easily decomposable carbon component in soil organic matter) was positively correlated with the production of $CH_4$ in an experiment on 23 soil samples in paddy fields from four different, southeast Asian countries [10]. This result indicated the importance of soil organic carbon, and indeed, soil organic carbon is a significant factor controlling $CH_4$ production in paddy fields. Sass et al. showed that $CH_4$ emissions during the growing season in paddy fields with different soil textures were positively correlated to the percentage of sand ($R^2 = 0.999$) during a four-year study, and indicated that $CH_4$ production was higher in sandy soils than clayey soils when the same amount of organic carbon was applied to the field [11]. Wang et al. observed the relationship between $CH_4$ emission and soil physicochemical properties of 16 paddy fields from USA, India, Thailand, and Liberia in an incubation experiment, and revealed that the decrease in redox potential (Eh) was correlated with biologically reducible Fe, Mn, and soil pH [12]. Moreover, Mitra et al. showed that Eh was correlated with cation exchange capacity (CEC) and available potassium [13].

Even though the knowledge of processes that contributes to the $CH_4$ emission is well-reported academically, understanding of upscaling or spatial level of the emission is still inadequate [14]. Moreover, The U.S. Environmental Protection Agency (USEPA) reported in 2006 that the increasing world population affects the demand of rice which lays an impact on methane emission. Specifically, three-fourths of the emission is emitted from south east Asian countries [15]. This means the assessment of methane production potential is highly essential as it implies how soil reduction processes perform [16]. With the reliable $CH_4$ production equation that is put into potentiality and represented in spatial level, the result can imply the possibility of a future amount of $CH_4$ emitted from paddy fields in wide scale.

Therefore, by determining the major soil properties that dominate $CH_4$ production, a strategy for soil amendment can be determined to reduce $CH_4$ emissions from paddy fields. In Thailand, the soil texture of major rice paddies ranges widely from sandy to clayey. In this study, topsoil collected from paddy fields in the central and northeastern parts of Thailand were used to measure $CH_4$ production potential ($CH_4PP$) in an anaerobic incubation experiment, and multiple regression models with several soil properties were

made. By interrogating the regression model, the regional difference in $CH_4PP$ between sandy and clayey soil areas was evaluated to investigate the effect of soil properties on $CH_4$ production in paddy soil.

## 2. Materials and Methods

### 2.1. Site Description

This study was conducted in the Ayutthaya province (14°21′6″ N, 100°34′38.6″ E), and in the Khonkaen (16°26′22.6500″ N, 102°49′43.4208″ E) and Mahasarakam provinces (16°11′05″ N, 103°18′02″ E) (Figure 1). Soil samples were randomly collected from a total of 44 paddy fields during June 2018, of which 20 samples (AY1 to AY20) were from Ayutthaya, and 24 samples (KK1 to KK24) were from Khonkaen and Mahasarakam (Table 1). Approximately 1 kg of composite soil from a depth of 0–15 cm was taken from five places in each field, mixed well, sealed in a plastic bag, brought back to the laboratory, air-dried, and used for the experiment. Separately, soil was sampled from a depth of 10 cm for the measurement of bulk density, and undisturbed core soil was sampled with a 100 mL stainless steel cylindrical tube.

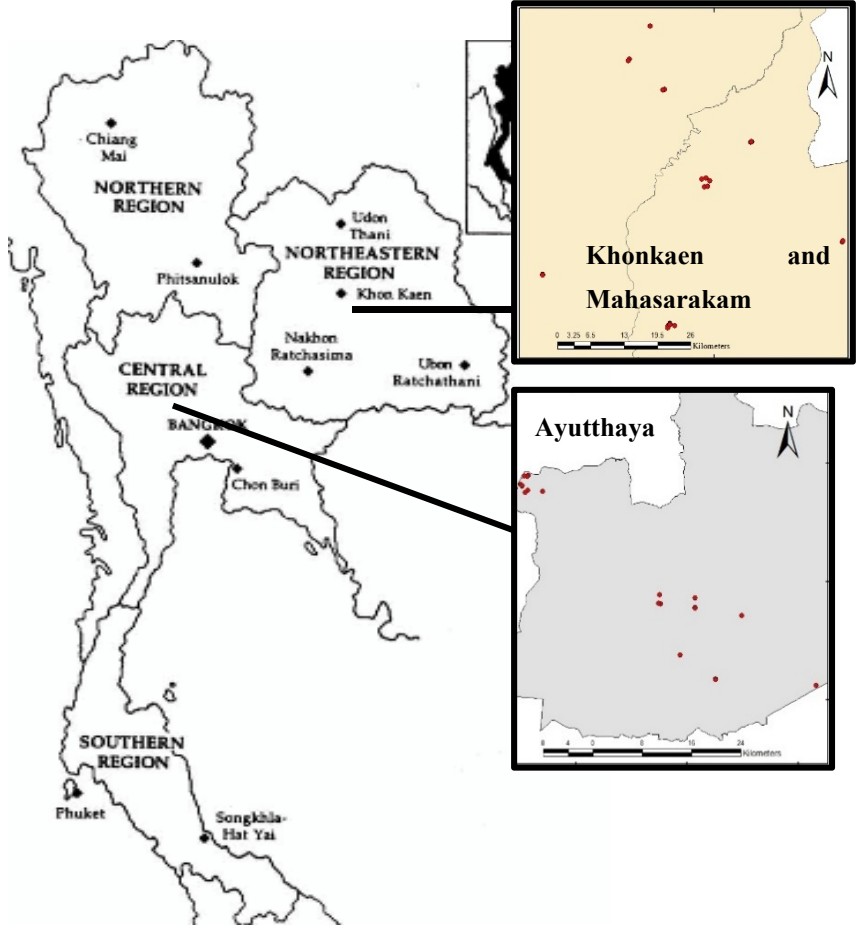

**Figure 1.** Study sites—Khonkaen, Mahasarakam, and Ayutthaya, Thailand.

**Table 1.** Sampling sites from Khonkaen, Mahasarakam, and Ayutthaya.

| Khonkaen and Mahasarakam | | Ayutthaya | |
| --- | --- | --- | --- |
| Site No. | Location | Site No. | Location |
| KK1 | 16°10′50.142″ N, 102°41′57.3396″ E | AY1 | 14°11′12.48″ N, 100°41′41.28″ E |
| KK2 | 16°10′51.9024″ N, 102°41′56.6268″ E | AY2 | 14°11′45.3372″ N, 100°32′36.7764″ E |
| KK3 | 16°32′38.4072″ N, 102°50′55.8384″ E | AY3 | 14°18′40.68″ N, 100°30′48.24″ E |
| KK4 | 16°32′47.0112″ N, 102°51′2.6712″ E | AY4 | 14°27′41.76″ N, 100°15′32.76″ E |
| KK5 | 16°36′9.8244″ N, 102°53′13.6356″ E | AY5 | 14°27′51.84″ N, 100°15′46.08″ E |
| KK6 | 16°29′38.4612″ N, 102°54′34.8696″ E | AY6 | 14°28′16.68″ N, 100°15′16.56″ E |
| KK7 | 16°29′40.9776″ N, 102°54′44.46″ E | AY7 | 14°28′14.52″ N, 100°15′16.56″ E |
| KK8 | 16°19′48.9864″ N, 102°59′19.068″ E | AY8 | 14°17′8.1816″ N, 100°35′1.3992″ E |
| KK9 | 16°19′47.424″ N, 102°58′56.8992″ E | AY9 | 14°29′6.72″ N, 100°15′28.44″ E |
| KK10 | 16°20′39.4728″ N, 102°59′6.1764″ E | AY10 | 14°28′22.08″ N, 100°15′7.92″ E |
| KK11 | 16°20′33.0396″ N, 102°58′39.6048″ E | AY11 | 14°29′3.84″ N, 100°15′45″ E |
| KK12 | 16°20′24.5868″ N, 102°59′31.9704″ E | AY12 | 14°29′7.8″ N, 100°15′49.32″ E |
| KK13 | 16°5′41.4924″ N, 102°55′48.4356″ E | AY13 | 14°27′47.16″ N, 100°17′7.8″ E |
| KK14 | 16°5′47.2848″ N, 102°55′18.7572″ E | AY14 | 14°11′47.04″ N, 100°32′36.96″ E |
| KK15 | 16°5′49.56″ N, 102°55′18.9264″ E | AY15 | 14°18′14.76″ N, 100°27′32.04″ E |
| KK16 | 16°5′52.746″ N, 102°55′19.7364″ E | AY16 | 14°18′56.88″ N, 100°27′36.72″ E |
| KK17 | 16°5′27.0492″ N, 102°55′4.9908″ E | AY17 | 14°18′11.16″ N, 100°27′41.76″ E |
| KK18 | 16°5′39.4548″ N, 102°55′9.0732″ E | AY18 | 14°17′49.2″ N, 100°30′50.04″ E |
| KK19 | 16°24′21.294″ N, 103°3′52.5852″ E | AY19 | 14°17′52.08″ N, 100°30′48.6″ E |
| KK20 | 16°24′19.8252″ N, 103°3′50.8968″ E | AY20 | 14°13′50.52″ N, 100°29′25.44″ E |
| KK21 | 16°24′23.4972″ N, 103°3′58.3884″ E | | |
| KK22 | 16°14′19.1796″ N, 103°13′29.9748″ E | | |
| KK23 | 16°14′16.5192″ N, 103°13′31.6776″ E | | |
| KK24 | 16°14′12.6996″ N, 103°13′29.406″ E | | |

Ayutthaya province has an area of 2556.64 km² in total and is in the central plain of Thailand. It is surrounded by rivers (JICA, 2007) and has mean annual precipitation, annual raining days, and maximum rain amount of 1023.7 mm·year⁻¹, 107 days, and 60.6 mm, respectively [17]. Khonkaen province is in a mountainous area, covering a total area of 10,886 km² [18] with a mean annual precipitation of 1304 mm·year⁻¹, total raining days of 112 days, and 64.4 mm maximum rainfall [17]. Mahasarakam province is next to Khonkaen province and has an area of 5292 km² with 1225.1 mm of annual rainfall,

102 raining days per year, and a maximum rainfall of 65.7 mm [19]. Based on soil texture analysis of the samples using the pipette method [20], Ayutthaya has clayey soil, and Khonkaen and Mahasarakam have sandy soil. For the purposes of classification of soil texture and location, Khonkaen and Mahasarakam were amalgamated as one province named Khonkaen. Sampling was conducted in June 2018.

### 2.2. Anaerobic Incubation Experiment

A 15 g sample of air-dried soil and 30 mL of deionized water were placed in a 100 mL bottle, 4 cm in diameter and 12 cm in height, following which the headspace was purged using $N_2$ (Hokkaido Air Water Inc., Hokkaido, Japan) to create anaerobic conditions (Figure 2). Two replicates were incubated during a period of 2 months at a temperature of 25 °C. During the incubation period, $CH_4$ was measured weekly using gas chromatography with a flame ionization detector (GC-14B, Shimadzu, Kyoto, Japan), and the vial bottles were flushed with $N_2$ after each gas detection to retain anoxic conditions. $CH_4$ production potential was calculated from fluctuation of gas concentration in the incubation bottle as follows in Formulas (1)–(3):

$$F = \rho \times (\text{gas concentration} \times V)/D \times W) \times \alpha \times 1000 \tag{1}$$

where, F is the gas emission($mg \cdot C \cdot kg^{-1} \cdot day^{-1}$); $\rho$ is the density of gas at the standard condition ($CH_4 = 0.717$ kg m$^{-3}$); gas concentration (ppmv); V(m$^3$) is volume of the bottle; W(g) is dry soil weight; $\alpha$ = is the conversion of factor for $CH_4$ to C (12/16). To calculate average cumulative $CH_4$ emission, the gas emission was utilized in the formular below:

$$\text{Average cumulative CH}_4 \text{ emission } \left(gC \cdot kg^{-1}\right) = F + (C \times D) \tag{2}$$

where, C is the last cumulative gas result; D is the number of days in the sampling interval.

$$CH_4 \text{ production potential (CH}_4\text{PP)} = \frac{\text{Average cumulative CH}_4\text{emission }\left(mgC \cdot kg^{-1}\right)}{\text{Number of incubation week}} \tag{3}$$

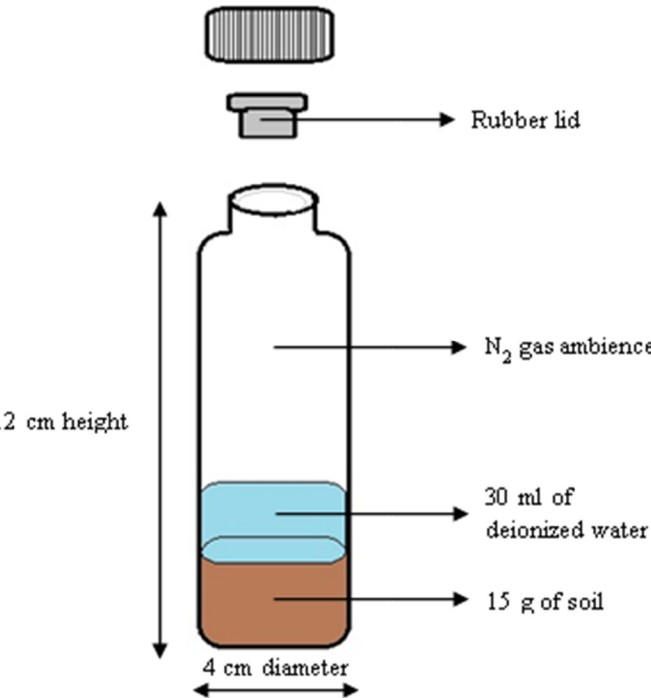

**Figure 2.** Anaerobic incubation.

### 2.3. Soil Properties Analysis

Soil was analyzed before and after incubation. Before incubation (BI), air-dried soil and deionized water were placed in a plastic container for the recording of 1:10 $pH_{BI}$ using a pH meter (pH meter F-22, Horiba, Kyoto, Japan), and electrical conductivity (1:10 $EC_{BI}$) using a conductivity meter (CM-30V, TOA-DKK). After filtration of the 1:10 water suspension, water-soluble anions ($SO_4^{2-}$, $NO_3^-$, $Cl^-$, and $PO_4^{3-}$) were analyzed by using ion chromatography (C-R8A chromato-processor, Shimadzu, Kyoto, Japan). Cation exchange capacity (CEC), exchangeable base cations (ExCa, ExMg, ExK, and ExNa) and base saturation (BS) were measured by the Schollenberger method [21]. Exchangeable base cations were examined using an atomic absorption spectrophotometer (200 Series AA, Agilent Technologies, Malaysia). Exchangeable $NH_4$ ($ExNH_4$(b)), extracted by 1 M KCl, was analyzed by the colorimetric method (Bolleter W. T., 1961) using a UV-VIS spectrophotometer (Shimadzu UV-1280, Shimadzu, Kyoto, Japan). Available phosphorus was analyzed using spectrophotometry (Spectronic Genesys 20, Becthai Bangkok Equipment & Chemical Co., Ltd., Bangkok, Thailand). Soil organic carbon (SOC) was obtained by the Walkley-Black acid digestion method (Walkey A.; Black, 1943), and the content of soil organic matter (SOM) was estimated by multiplying SOC content by 1.72. Total nitrogen was determined by the Kjeldahl method [22] (Table S2 in supplementary).

After incubation (AI), soil samples were moved to a plastic container, and 120 mL of deionized water were added, after which 1:10 $pH_{AI}$ and 1:10 $EC_{AI}$ were measured. The suspension was then filtered, and water-soluble anions, water-soluble organic carbon (WSOC), and inorganic carbon (IC) were analyzed using a TOC analyzer (TOC-5000A, Shimadzu, Kyoto, Japan). Soil samples were further extracted with 120 mL of 1M KCl to measure exchangeable ammonium ($ExNH_{4(AI)}$), ferrous iron ($Fe^{2+}$), and manganese ($Mn^{2+}$). $ExNH_4$ and $Fe^{2+}$ were analyzed by colorimetric methods with a UV-VIS spectrophotometer (Shimadzu UV-1280, Shimadzu, Kyoto, Japan), and $Mn^{2+}$ was analyzed by using an atomic absorption spectrophotometer (Z-5010, Hitachi, Tokyo, Japan).

### 2.4. Regression Model for $CH_4PP$

Significant factors controlling $CH_4PP$ were analyzed using the Pearson's correlation matrix and a step-wise multiple regression analysis in IBM SPSS Statistics for Windows, version 21 [23]. The comparison of variable means between the areas was determined using one-way ANOVA at 95% confidential level.

Multicollinearity usually occurs with a number of independent values that are highly correlated. To overcome this problem, one each of the highly correlated paired independent parameters (at $R^2 > 0.7$) was removed, and the variance inflation factor (VIF) was calculated for the remaining parameters. Multicollinearity was prevented in independent variables with a VIF of less than 10 [24]. Therefore, the variables with VIF less than 10 were included in the stepwise regression to predict $CH_4PP$.

### 2.5. Evaluation of the Regional Differences in $CH_4$ Production Potential

In order to compare the $CH_4$ production potential between the Ayutthaya clayey soil area and the Khonkaen sandy soil area, spatial $CH_4$ production potential was predicted according to Figure 3. Two experimental area map datasets and one in vitro predicted soil dataset were employed in order to invent predicted regional $CH_4PP$ map: (1) the soil properties map in point data was provided by the Land Development Department (LDD) of Thailand [25] consisting of organic matter (%), 1:1 pH, organic carbon (%), available phosphorus ($mg \cdot kg^{-1}$), and exchangeable potassium ($mg \cdot kg^{-1}$), and (2) Online OpenLandMap from GitHub (© LandGIS contributors, n.d.) in a raster data structure that represented sand content (%), and clay content (%). Ayutthaya and Khonkaen map datasets consisted of 1490 points and 4253 points, respectively. Because the map dataset of GitHub was in a raster data structure, the datasets were interpreted in ArcGIS Desktop version 10.1 [26].

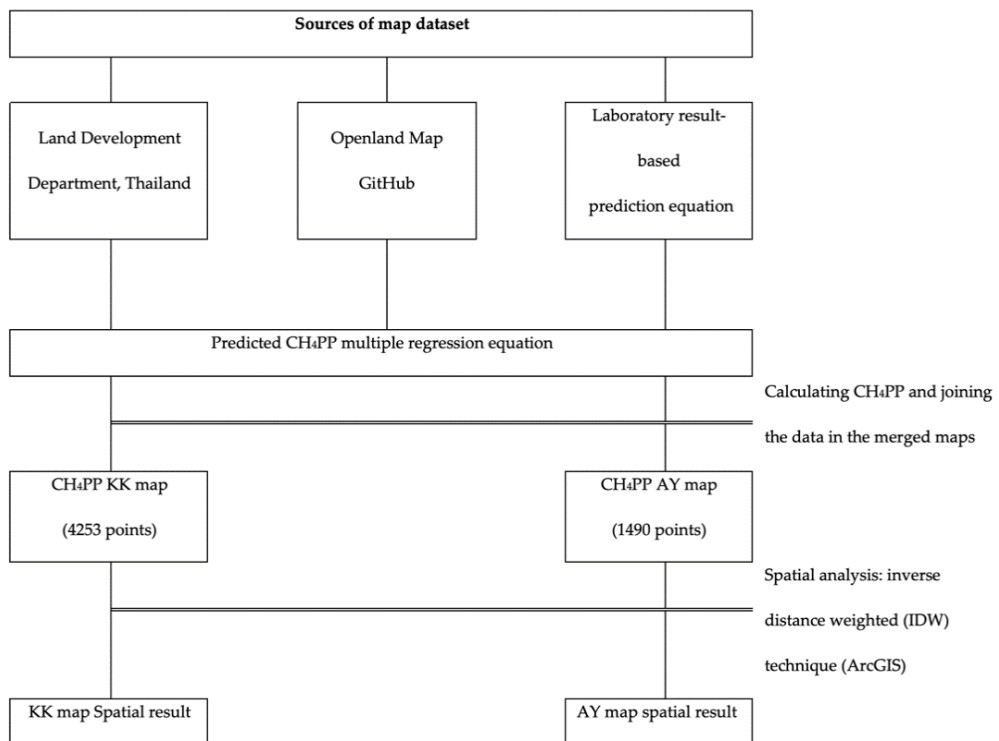

**Figure 3.** Workflow schematic of map spatial analysis.

The multiple regression analysis for $CH_4PP$ showed that $CH_4PP$ was significantly correlated with SOC, 1:10 $EC_{BI}$, $ExNH_{4(BI)}$, and sand content. However, 1:10 $EC_{BI}$ and $ExNH_{4(BI)}$ were not available in LDD. Therefore, regression equations were produced to predict those parameters using the available soil properties in LDD. The equations were obtained as follows in Formula (4) and (5):

$$ExNH_{4(BI)}\left(mg \cdot kg^{-1}\right) = 0.19 Silt(\%) + 6.2 (R^2 = 0.17, \ p < 0.01) \tag{4}$$

$$1:10 \ ECBI\left(mS \cdot m^{-1}\right) = 3.1 \ SOM(\%) - 5.03 pH_{BI} + 33.56 \ (R^2 = 0.74, \ p < 0.01) \tag{5}$$

The $CH_4PP$ prediction equation was used to predict $CH_4$ production potential at the sampling points in each province. An inverse distance weighted technique (IDW) was run in ArcGIS 10.1 for regional $CH_4$ production potential in the provinces. IDW was also appointed to display the significant soil properties of the areas spatially.

## 3. Results

As shown in Figure 4a, there was no significant difference in the average $CH_4PP$ between Ayutthaya and Khonkaen (2012.31 and 1742.81 mg·C·kg$^{-1}$·week$^{-1}$, respectively, $p > 0.05$). For the soil properties measured before incubation, Ayutthaya was significantly greater than Khonkaen ($p \leq 0.05$) in 1:10 $EC_{BI}$ (10.69 and 1.97 mS.m$^{-1}$), SOM (4.26% and 1.27%), SOC (2.46% and 0.74%), $ExNH_{4(BI)}$ (17.07 and 12.38 mg·kg$^{-1}$), ExK, ExCa, ExMg, ExNa, Total N, CEC, $SO_4{}^{2-}$-$S_{BI}$, and silt. On the other hand, Ayutthaya was significantly lower than Khonkaen in 1:10 $pH_{BI}$ (6.08 and 6.75), sand (4.38% and 44.62%) and $NO_3{}^-$-$N_{BI}$. For the soil characteristics measured after incubation, Ayutthaya was significantly higher than Khonkaen in 1:10 $EC_{AI}$, $SO_4{}^{2-}$-$S_{AI}$, $ExNH_{4(AI)}$ and IC, while $Mn^{2+}$ was significantly lower in Ayutthaya than in Khonkaen (Figure 5). There was no significant difference between Ayutthaya and Khonkaen in $Fe^{2+}$ and WSOC (Figure 5c,j).

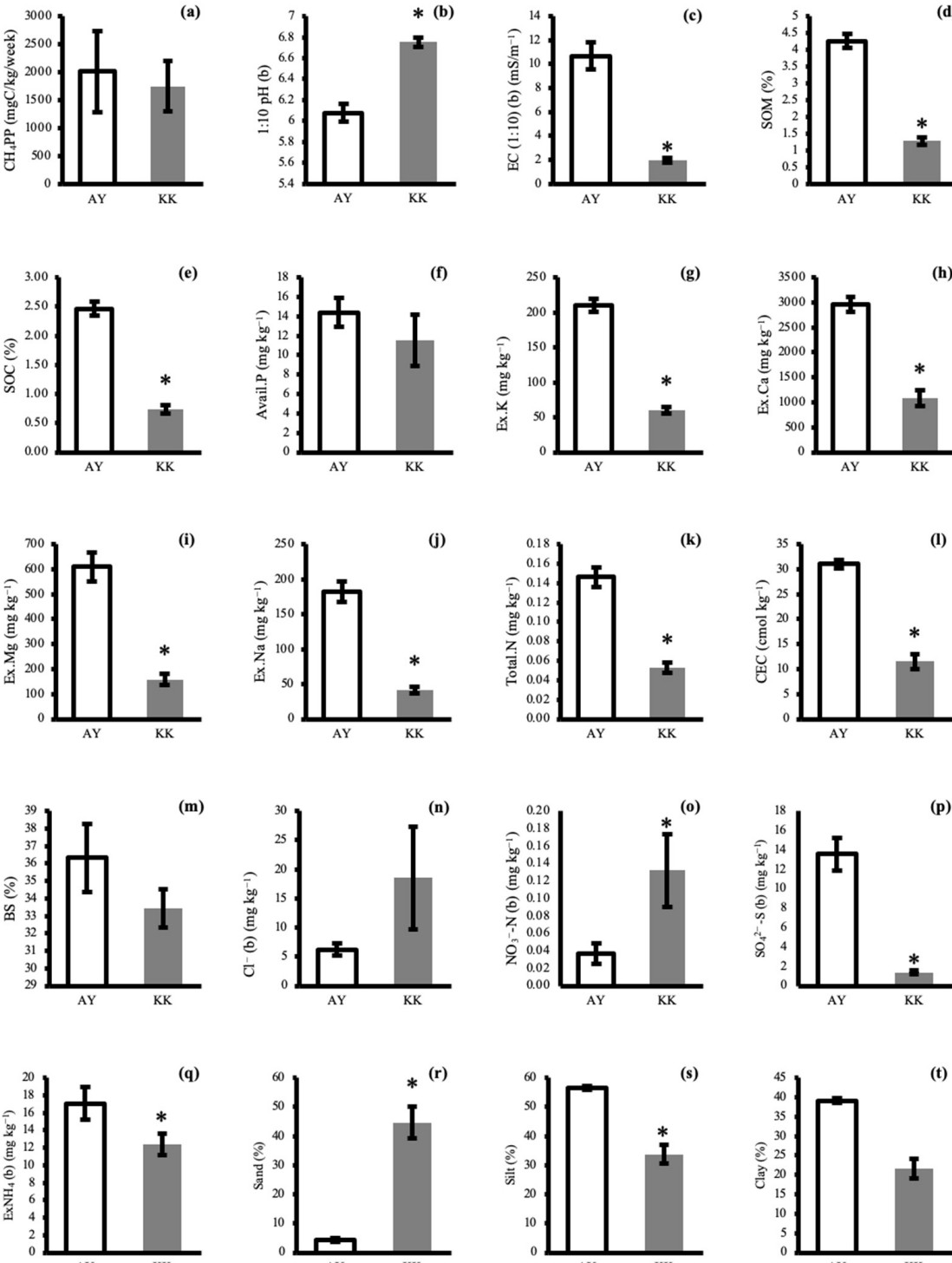

**Figure 4.** A comparison of average differences in $CH_4$ production potential and soil properties before incubation in Ayutthaya and Khonkaen. $CH_4PP$ = $CH_4$ production potential (**a**); 1:10 pH(b) (**b**); 1:10 EC(b) = 1:10 electronic conductivity (**c**); SOM = Soil organic matter (**d**); SOC = Soil organic carbon (**e**); Avail.P = Available phosphorus (**f**); Ex.K = exchangeable potassium (**g**); Ex.Ca = Exchangeable calcium (**h**); Ex.Mg = Exchangeable magnesium (**i**); Ex.Na = Exchangeable sodium (**j**); Total.N = Total nitrogen (**k**); CEC = Cations exchange capacity (**l**); BS = Base saturation (**m**); $Cl^-$ = Chloride (**n**); $NO_3^-$-N = Nitrate (**o**); $SO_4^{2-}$-S = Sulfate (**p**); $ExNH_4$ = Exchangeable ammonium (**q**); Sand(%) (**r**); Silt(%) (**s**); Clay(%) (**t**). Asterisk (*) represents a significant difference at the $p < 0.05$ level between the 2 areas. $PO_4^{3-}$-P is not shown.

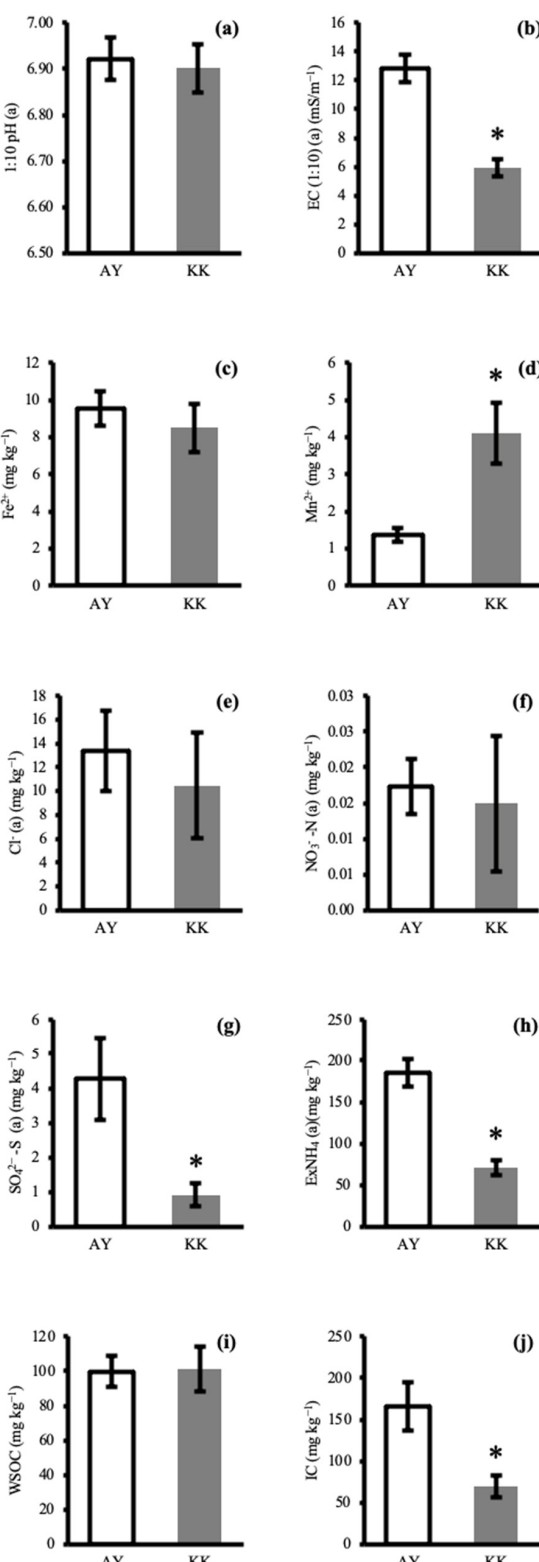

**Figure 5.** A comparison of average differences in soil properties after incubation in Ayutthaya and Khonkaen. 1:10 pH(a) (**a**); 1:10 EC(a) = 1:10 electronic conductivity (**b**); $Fe^{2+}$ = Ferrous ion (**c**); $Mn^{2+}$ = Manganese ion(**d**); $Cl^-$ = Chloride (**e**); $NO_3^--N$ = Nitrate (**f**); $SO_4^2—S$ = Sulfate (**g**); $ExNH_4$ = Exchangeable ammonium (**h**); WSOC = Water soluble organic carbon (**i**); IC = Inorganic carbon (**j**). Asterisk (*) represents a significant difference at the $p < 0.05$ level between the 2 areas. Figures of $PO_4^{3-}-P$ and $NO_2^--N$ are not shown.

Soil properties after incubation were higher than before incubation in 1:10 EC (9.06 and 5.9 mS·m$^{-1}$, respectively), 1:10 pH, ExNH$_4$. In contrast, anions after incubation were lower than before incubation in PO$_4^-$, SO$_4^{2-}$, NO$_3^-$, and Cl$^-$.

Results of the simple regression analysis showed that although there was no significant correlation between CH$_4$PP and each soil property measured before incubation, except for ExNH$_{4(BI)}$ (Table S3), CH$_4$PP was significantly correlated with 1:10 pH$_{AI}$, Fe$^{2+}$, and WSOC measured after incubation (Table 2). Using the soil characteristics measured before incubation, the result of step-wise multiple regression analysis showed that CH$_4$PP was significantly correlated with 1:10 EC$_{BI}$, SOC, ExNH$_{4(BI)}$, and sand content (Figure 6), and the multiple regression equation indicated that all of the soil properties except for 1:10 EC$_{BI}$ were positively correlated with CH$_4$PP without multicollinearity due to a VIF < 10 (Table 3). After incubation, however, the result of multiple regression analysis showed that only 1:10 pH$_{AI}$ was significant (R$^2$ = 0.22, $p$ < 0.01).

**Table 2.** Correlation matrix among CH$_4$ Production Potential (CH$_4$PP) and soil properties (after incubation).

| | CH$_4$PP | pH (AI) | EC(AI) | Fe$^{2+}$ | Mn$^{2+}$ | Cl$^-$ (AI) | NO$_3^-$ (AI) | SO$_4^{2-}$ (AI) | ExNH$_4$(AI) | TOC | IC |
|---|---|---|---|---|---|---|---|---|---|---|---|
| CH$_4$PP | | | | | | | | | | | |
| pH$_{AI}$ | −0.49 ** | | | | | | | | | | |
| EC $_{AI}$ | −0.07 | 0.02 | | | | | | | | | |
| Fe$^{2+}$ | 0.30 * | −0.49 | −0.15 | | | | | | | | |
| Mn$^{2+}$ | −0.17 | 0.23 | −0.16 | −0.28 | | | | | | | |
| Cl$^-$ $_{AI}$ | −0.16 | 0.16 | 0.26 | −0.2 | 0.01 | | | | | | |
| NO$_3^-$ $_{(AI)}$ | −0.13 | −0.01 | 0.06 | −0.05 | 0.04 | 0.03 | | | | | |
| SO$_4^{2-}$ $_{(AI)}$ | −0.02 | −0.25 | 0.69 ** | −0.02 | −0.21 | 0.04 | −0.02 | | | | |
| ExNH$_4$ $_{(AI)}$ | 0.07 | −0.12 | 0.46 * | 0.16 | −0.19 | 0.02 | 0.12 | 0.23 | | | |
| WSOC | 0.35 * | −0.31 | −0.01 | 0.13 | −0.25 | −0.15 | −0.19 | 0.99 | −0.17 | | |
| IC | −0.25 | 0.42 ** | 0.32 * | −0.21 | −0.15 | 0.317 * | 0.25 | −0.08 | 0.55 ** | −0.29 | |

* Correlation is significant at the 0.05 level (2-tailed). ** Correlation is significant at the 0.01 level (2-tailed).

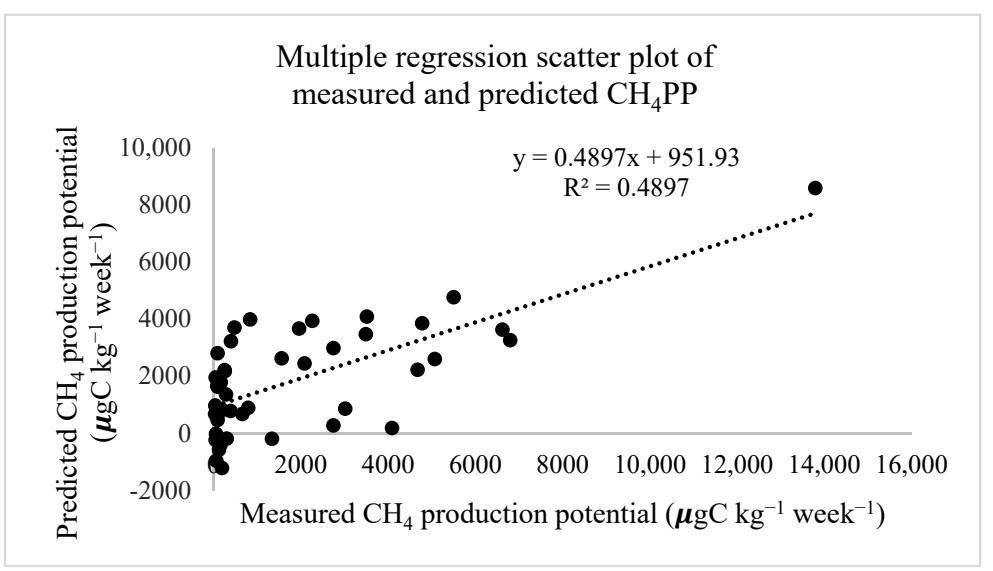

**Figure 6.** Multiple regression scatter plot between CH$_4$ production potential (CH$_4$PP) and major soil properties before incubation.

**Table 3.** Significant predictors in the regression before incubation (BI).

| Predictors | Beta | Significant | Regression Equation | $R^2$ ($p$ Value) | VIF |
|---|---|---|---|---|---|
| SOC (%) | 1.3 | 0.00003 | CH$_4$PP($\mu$g C/kg/week) = 3649.1(SOC) + 120.7(ExNH$_{4(BI)}$) + 89.5(Sand) $-$ 328.5(1:10 EC$_{BI}$) $-$ 5845.7 | 0.50 (0.00002) | 8.3 |
| ExNH$_{4(BI)}$ (mg$\cdot$kg$^{-1}$) | 0.3 | 0.01 | | | 1.3 |
| Sand (%) | 0.9 | 0.00005 | | | 4.7 |
| 1:10 EC$_{BI}$ (mS$\cdot$m$^{-1}$) | $-0.7$ | 0.002 | | | 3.3 |

The regional CH$_4$PP predicted by the multiple regression equation using the map parameter values was not different between Ayutthaya and Khonkaen (Table 4). The differences in SOC, ExNH$_{4(BI)}$ and 1:10 EC$_{BI}$ were significantly larger in Ayutthaya than in Khonkaen. Consistently, the sand content in Khonkaen was more than twice that of Ayutthaya (49.76 and 21.01%). In Ayutthaya, 1490 variables were used, and 4253 variables were used in Khonkaen.

**Table 4.** Map datasets: Average and standard deviation of CH$_4$ production potential (CH$_4$PP) and major soil properties and significant level for the difference between Ayutthaya and Khonkaen by ANOVA.

| Variables | Ayutthaya | Khonkaen | $p$ Value |
|---|---|---|---|
| CH$_4$PP ($\mu$g$\cdot$Ckg$^{-1}\cdot$week$^{-1}$) | 329.9 $\pm$ 956.32 | 370.37 $\pm$ 1061.15 | 0.19 |
| SOC (%) | 1.54 $\pm$ 0.68 | 0.46 $\pm$ 0.47 | <0.001 |
| ExNH$_{4(BI)}$ (mg$\cdot$kg$^{-1}$) | 12.60 $\pm$ 0.48 | 11.23 $\pm$ 0.52 | <0.001 |
| Sand (%) | 21.01 $\pm$ 6.03 | 49.76 $\pm$ 6.63 | <0.001 |
| 1:10 EC$_{BI}$ (mS$\cdot$m$^{-1}$) | 13.5 $\pm$ 4.25 | 7.41 $\pm$ 4.06 | <0.001 |

The distribution map of predicted CH$_4$PP and the soil properties as the significant controlling factors of CH$_4$PP in Ayutthaya and Khonkaen are shown in Figures 7 and 8, respectively. The distribution patterns of CH$_4$PP corresponded well to that of SOC in both regions (Figure 7a,b and Figure 8a,b), but corresponded less to ExNH$_{4(BI)}$ in Khonkaen than Ayutthaya (Figures 7c and 8c). There was an inverse relationship between 1:10 EC$_{BI}$ and CH$_4$PP (Figures 7d and 8d), especially in the area of low sand content (Figures 7e and 8e).

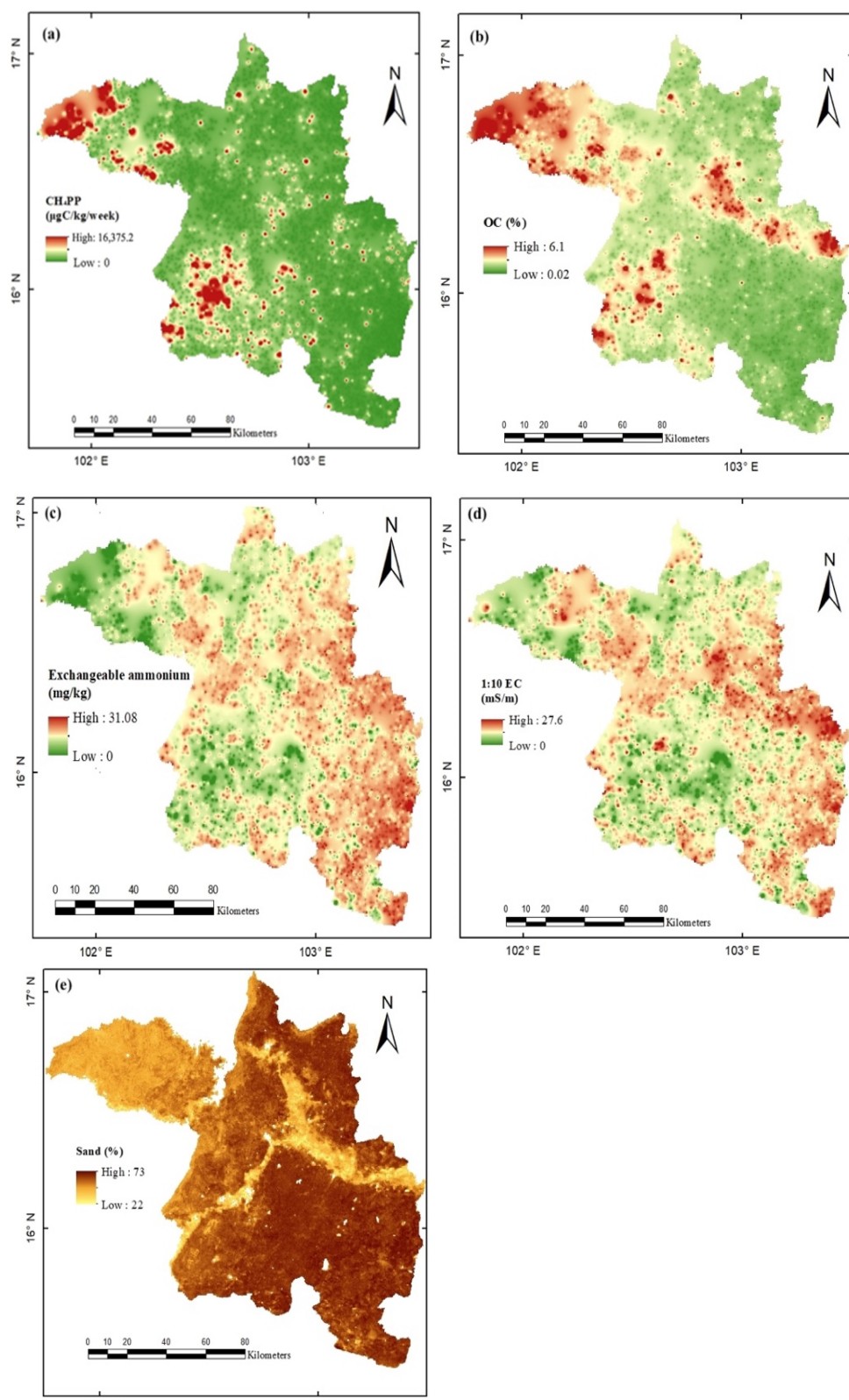

**Figure 7.** Spatial map of predicted CH$_4$ production potential (**a**), soil organic carbon (**b**), predicted exchangeable ammonium (**c**), 1:10 predicted electrical conductivity (**d**), and sand content (**e**) in Khonkaen province.

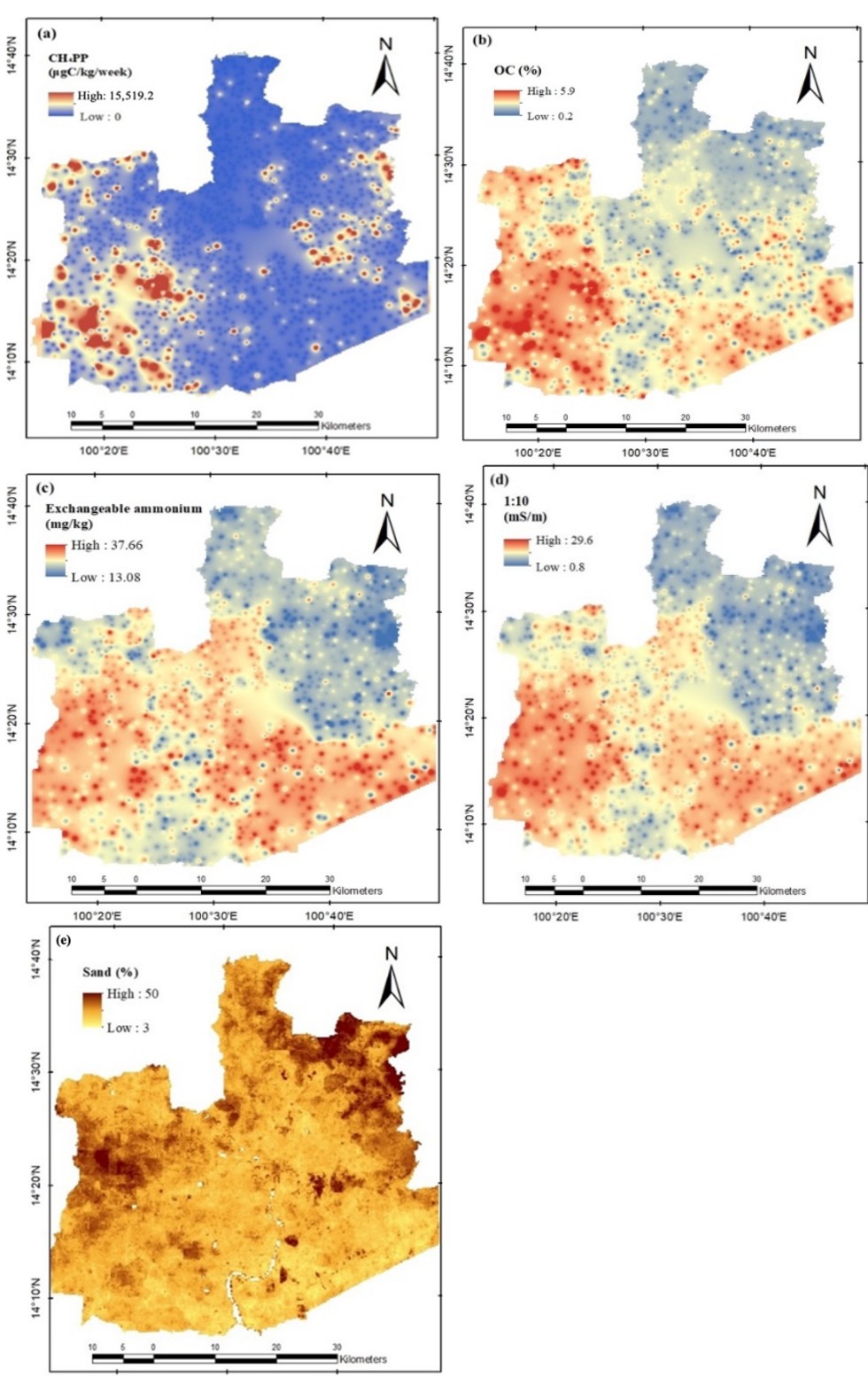

**Figure 8.** Spatial map of predicted $CH_4$ production potential (**a**), soil organic carbon (**b**), predicted exchangeable ammonium (**c**), 1:10 predicted electrical conductivity (**d**), and sand content (**e**) in Ayutthaya province.

## 4. Discussion

Sand content has been suggested as a significant factor controlling $CH_4$ emissions because of the low oxide content [11,27]. However, SOC and other soil properties re-

lated to redox potential have also been reported as significant factors controlling $CH_4$ emissions [28,29]. In the current study, the mean value of $CH_4PP$ and more than half of the soil properties were statistically higher in Ayutthaya clayey soil than in Khonkaen sandy soil (Figures 4 and 5). Therefore, soil properties is significantly related to $CH_4$ emission [3]. Soil properties measured before incubation did not correlate with $CH_4PP$ in simple regression (Table S3). A regression equation was obtained by stepwise multiple regression analysis relating $CH_4PP$ to the soil properties (Table 3). In the regression equation, SOC, sand, and $ExNH_{4(BI)}$ contributed positively to $CH_4PP$, but 1:10 $EC_{BI}$ contributed negatively to $CH_4PP$ (Table 3). Within the soil properties measured after incubation, $CH_4PP$ showed significant negative correlations with 1:10 $pH_{AI}$, and a significant positive correlation with $Fe^{2+}$ and WSOC (Table 2). These observations are reasonable because the rise in pH indicates the presence of free iron, which suppresses $CH_4$ production. Increases in WSOC provide evidence of the decomposition of OC, leading to the production of $CH_4$. Moreover, the incline of $Fe^{2+}$ implies the high reduction of $Fe^{3+}$, which organic carbon is acting as electron donor causing $CH_4$ production. On average, of the properties 1:10 EC, 1:10 pH, $ExNH_4$, and anions ($PO_4^-$, $SO_4^{2-}$, $NO_3^-$, $Cl^-$) measured before incubation, 1:10 EC, 1:10 pH, and $ExNH_4$ increased, but anions decreased after incubation.

The SOC in submerged soil acts as an electron donor in the reduction process, enabling methanogens to generate $CH_4$ by decomposing SOC [3]. It is well known that direct incorporation of rice straw into paddy field soils increases $CH_4$ significantly, and the application of cow manure contributes to higher $CH_4$ emission than inorganic fertilizer only [19,30,31]. Chidthaisong et al. observed the response in $CH_4$ production under anaerobic incubation to different fertilizer inputs, and found that $CH_4$ formation was the highest for plots with cow manure, following by rice straw, rice straw with chemical fertilizer, and chemical fertilizer alone [32]. These decomposed organic carbons can not only perform as electron donor to generate $CH_4$ production, but also can induce $Fe^{2+}$ production which can be speculated from increasing amount of $Fe^{3+}$ reduction [33].

Sass et al. observed $CH_4$ emissions in paddy fields in a wide range of soil textures and reported that an increase in sand content elevated $CH_4$ emission in anaerobic conditions [11]. In a comparison of flooded fields of clayey soil and sandy soils with the same organic carbon application, sandy soil emitted more $CH_4$ than a clayey soil [27]. Following oxygen depletion, when $CH_4$ is produced, the sequence of electron acceptor use is $NO_3^-$, $MnO_2$, $Fe_2O_3$, $SO_4^{2-}$ and $CO_2$, where organic matter acts as the electron donor [9]. Usually, sandy soil has a lower amount of free $MnO_2$ and $Fe_2O_3$ as compared to clayey soil [34]. Thus, the reduction process occurs more quickly in sandy soil than in clayey soil. In the current study, $CH_4PP$ was similar between Ayutthaya and Khonkaen, and $Fe^{2+}$ production was not significantly different between Ayutthaya and Khonkaen (Figure 5c), and a higher amount of $Mn^{2+}$ production was shown in Khonkaen than in Ayutthaya (Figure 5d). On the other hand, $SO_4^{2-}$ was reduced more in Ayutthaya than in Khonkaen. Therefore, $SO_4^{2-}$ reduction might be a main suppressor of $CH_4PP$ in this study.

The electrical conductivity of soil water extraction is a significant indicator of water-soluble ion concentrations [35]. In the present study, only $SO_4^{2-}$ correlated significantly with 1:10 EC (Table 2). Figure 9 shows a positive relationship between 1:10 $EC_{BI}$ and $SO_4^{2-}{}_{BI}$ ($R^2 = 0.34$, $p = 0.00003$). As $SO_4^{2-}$ reduction would occur immediately before the production of $CH_4$ [9], this clearly supports the negative influence that 1:10 $EC_{BI}$ has on $CH_4PP$ as shown in the multiple regression (Table 3). The significant positive relationship between $ExNH_{4(BI)}$ and $CH_4PP$ may be due to $NH_4^+$ being the product of organic matter decomposition. This is supported by the significant positive relationship between WSOC and $CH_4PP$ (Table 2).

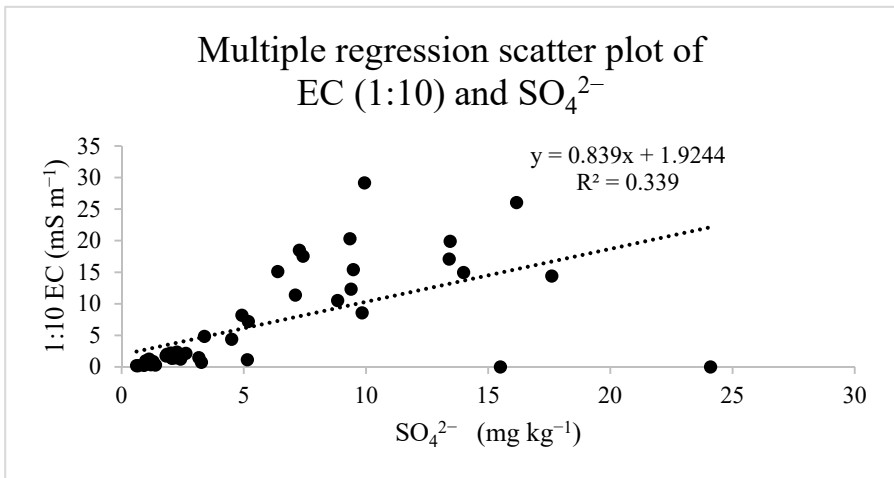

**Figure 9.** Multiple regression scatter plot between 1:10 ECBI and $SO_4^{2-}$ (BI = before incubation). 1:10 EC = 1:10 electronic conductivity; $SO_4^{2-}$ = Sulfate.

As shown in Figures 7 and 8, the distribution pattern of the predicted $CH_4PP$ was strongly influenced by SOC in Ayutthaya, and sand content in Khonkaen. Higher $CH_4PP$ tended to be shown at higher SOC content in Ayutthaya because it acts as electron donor which supports anaerobic reduction process. A large amount of $CH_4PP$ found in areas with high sand content in Khonkaen indicates that sandy soil that contain lowest free ion among other soils can perform poorly at slowing down the reduction process [11]. Moreover, in Ayutthaya, $CH_4PP$ tended to correspond to $ExNH_4$, which could be because $ExNH_4$ is a product of organic matter decomposition which organic matter is an important electron donor in soil anaerobic reduction providing energy for microbial metabolism. Elevated $CH_4PP$ occurred at low 1:10 $EC_{BI}$, but 1:10 $EC_{BI}$ in Ayutthaya fluctuated more than Khonkaen. This is because 1:10 $EC_{BI}$ was influenced by $SO_4^{2-}$ reduction, one of the reduced substrates in the process indicating that lower electron contained in soil would accelerate soil reduction anaerobically. Ayutthaya soil contained a wide range of $SO_4^{2-}$.

A digital soil map (DSM) attributes soil properties and provides pedological knowledge to users [36]. Pedotransfer functions (PTFs) provide a means of using little soil information to create a meaningful and larger soil data image in DSM. A prediction equation from stepwise regression was determined in the current study, and the result is a form of PTF, but improved, in that it works on the prediction process and simultaneously enables users to learn pedologically fundamental knowledge.

Uncertainty in mapping prediction is a threat and leads to invalid conclusions. To avoid uncertainty, sufficient predictors, and a high density of data points are essential. A reliable regression prediction, which implies a statistically acceptable relationship between predictors and the predicted value, is required [36]. Referring to Table 2 and Figure 3, our study utilized various soil properties as well as sufficient points from the soil database. In addition, using multiple regression (a PTF) as a tool to predict unavailable soil properties in map data with a 99.99% confidence level ($R^2 = 0.50$), with VIF less than 10 at all predictors (Table 3), meant that half of sampling results confidently met the prediction made in the course of this study.

## 5. Conclusions

This study demonstrated the importance of soil properties on $CH_4$ emission in Thai paddy fields. The spatial prediction result showed SOC, EC, $ExNH_4$, and sand content are $CH_4$ production influential factors in the study area. Moreover, our incubation experiment showed that $CH_4$ production in this study is influenced by SOC, EC, $ExNH_4$, sand content, $Fe^{2+}$, $SO_4^{2-}$, and pH. The above interaction of each soil property and $CH_4$ production can be explained as follows: Because SOC is a derived form of organic matter, the increases in WSOC provides evidence of the decomposition of SOC which is important as it acts

as electron donor in reduction process resulting the production of $CH_4$. Referring to the role of SOC as electron donor, $NH_4^+$ indirectly supports the $CH_4$ production because it is the product of organic matter decomposition. Furthermore, both $CH_4PP$ and $Fe^{2+}$ were found increasing simultaneously due to the increase of $Fe^{2+}$ which implies the increase of $Fe^{3+}$ reduction rate while additional SOC acting as an extra electron donor acceleratively causing $CH_4$ production. As sand content has the least iron oxide compared to other soil texture, it, therefore, leads to quick consumption of electron acceptors causing $CH_4$ production to increase. On the other hand, $SO_4^{2-}$ is one of electron acceptors reduced in reduction process and the reduction of $SO_4^{2-}$ implies the reduction of EC. This means the decrease of EC in soil increases $CH_4$ production. pH in soil implied the presence of iron oxide in soil. At higher pH, soil would attach more iron oxides which helps delaying reduction process which suppresses $CH_4$ production. To conclude, soil properties related to reduction reactions, as well as soil texture, are strongly influential in accounting for $CH_4PP$ in the paddy fields in the two regions in Thailand. Further than that, this study showed that the result of the multiple regression analysis exploring the soil properties that controls $CH_4$ production can be used as a pedotransfer function, which is able to predict a map of regional $CH_4$ production using the map data sets of soil properties.

**Supplementary Materials:** The following are available online at https://www.mdpi.com/article/10.3390/agriculture11050467/s1, Table S1: Details of reagents used in the study, Table S2: Soil characteristics and analysis method before and after soil incubation, Table S3: Correlation matrix for $CH_4$ production potential ($CH_4PP$) and soil properties (before incubation).

**Author Contributions:** Conceptualization, P.S., N.A., S.A., R.H.; methodology, P.S., S.A., R.H.; software, P.S., N.A.; validation, P.S. and R.H.; formal analysis, P.S., S.A.; investigation, P.S., N.A., S.A., R.H.; resources, P.S., N.A., S.A., R.H.; data curation, P.S., S.A.; writing—original draft preparation, P.S.; writing—review and editing, P.S., R.H.; visualization, P.S., N.A., S.A., R.H.; supervision, N.A., S.A., R.H.; project administration, P.S. and R.H.; funding acquisition, R.H. All authors have read and agreed to the published version of the manuscript.

**Funding:** This research was funded by Heiwa Nakajima Foundation, grant number 2018.

**Institutional Review Board Statement:** Not applicable.

**Informed Consent Statement:** Not applicable.

**Acknowledgments:** This research was conducted using International Joint Research Grant for "Assessing mitigation options reducing $CH_4$ emission from rice paddy fields in Thailand" by Heiwa Nakajima Foundation in 2018. Furthermore, we sincerely thank the Japan International Cooperation Agency (JICA) for providing financial support through the Asia Innovative Scholarship.

**Conflicts of Interest:** The authors declare no conflict of interest.

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
