# Peer review of "Evaluation of CH4 Emission in Two Paddy Field Areas, Khonkaen and Ayutthaya, in Thailand"

_agriculture, doi:10.3390/agriculture11050467_

Round 1

Reviewer 1 Report

Manuscript has been significantly improved by authors and therefore i recommend present manuscript for publication. 

Author Response

Thank you for your feedback. 

Reviewer 2 Report

Authors have addressed and/or clarified the comments and suggestions raised in the review. However, authors are yet to revise conclusion. Once it is revised, MS can be better. Minor language editing is required.

L13: replace parameter with 'properties'

L401: remove 'and soil texture'. texture comes under soil properties.

Check throughout the text and replace 'prameters' by properties'

L404: replace parameter with 'properties'

As suggested, conclusion is not revised. Please remove result part from conclusion. Focus on your findings and their implications, e.g., what are the importance of soil properties on emissions, spatial variabities etc rather than comparing two sites which is already done under discussion.

Author Response

Thank you for your suggestion.

L13: replace parameter with 'properties'

L401: remove 'and soil texture'. texture comes under soil properties.

Check throughout the text and replace 'prameters' by properties'

L404: replace parameter with 'properties'

The above parts were modified.

As suggested, conclusion is not revised. Please remove result part from conclusion. Focus on your findings and their implications, e.g., what are the importance of soil properties on emissions, spatial variabities etc rather than comparing two sites which is already done under discussion.

The conclusion were altered according to the suggestion.

I hope the modified version attached here finds you well.

This manuscript is a resubmission of an earlier submission. The following is a list of the peer review reports and author responses from that submission.

Round 1

Reviewer 1 Report

The manuscript submitted in Journal agriculture, entitled “Evaluation of CH4 emission in two paddy field areas, Khon-2 kaen and Ayutthaya, in Thailand” has been critically reviewed. Methane is second most potent greenhouse gas after carbon dioxide and its production mechanistic understating in soil is crucial as methane play significant role in global warming phenomena. Manuscript have high scientific sound but it need some revision, some of the suggestion for the authors are given below:  

General comment: In thinks authors forget to add all figures????? Kindly add all the figures in manuscript before submitting revision.   

Use full name before writing abbreviation, example line no 11, CH4,    

References citation need formatting as per journal guideline

Other Comments:   

Line no 35-36: Submerged sandy soils emit high levels of CH4 due to free iron oxides causing rapid reduction reactions- support with the relevant reference.

Line no 37, 41, Rice paddy: use either word rice or paddy instead for Rice paddy in manuscript

Line no 106, Section 2.2 I believe that, if authors add image of anaerobic incubation experiment in this section it will be great.

The real image provide better and clear cut information for new researcher who are willing to conducted similar experiment. (If you do not have real image kindly try to draw sketch of set)  

Line no 113 to 115- Kindly add details about calculation of CH4 production and its CH4PP in details (Provide formula used by authors).

Table 2: remove this table from main manuscript and add to supplementary file.

Author Response

Please also see the manuscript attached for more understanding to each reply.

And, the reply to each comment will be below.

General comment: In thinks authors forget to add all figures????? Kindly add all the figures in manuscript before submitting revision.   

Thank you. I separated the figures in another file. As your suggestion, I moved them and rearranged all figures in the manuscript.

Use full name before writing abbreviation, example line no 11, CH4,    

I followed your request as follows:

Abstract: It is well known that submerged soils emit high levels of methane (CH4) due to oxygen deprivation and free iron oxide causing a quick reduction.

References citation need formatting as per journal guideline

 Modification was made. Thank you for your suggestion.

Other Comments:   

Line no 35-36: Submerged sandy soils emit high levels of CH4 due to free iron oxides causing rapid reduction reactions- support with the relevant reference.

 Thank you. This comment was received as same as the other reviewer who recommended erasing “SANDY” from the part of article as it is more common to say “submerged soil” as follows:

In recent decades, causes and mitigation issues related to global warming have become controversial. Methane (CH4) is a stronger greenhouse gas (GHG) than CO2 because it has a  higher radiative trapping ability [13]. Submerged soils emit high levels of CH4 due to free iron oxides causing rapid reduction reactions.

Line no 37, 41, Rice paddy: use either word rice or paddy instead for Rice paddy in manuscript

Thank you. The part was modified as follows:

In agriculture, paddy fields are an important source of atmospheric CH4, as flooded conditions are preferable for proper rice growth [17,21]. Globally, rice cropping is considered to account for 5 to 20% of total CH4emission from anthropogenic actions [12]. Paddy fields in Thailand cover almost half the total agricultural land and are mostly located in the central and northeastern parts of the country.

Line no 106, Section 2.2 I believe that, if authors add image of anaerobic incubation experiment in this section it will be great.

The real image provide better and clear cut information for new researcher who are willing to conducted similar experiment. (If you do not have real image kindly try to draw sketch of set)  

Since I do not have a clear photo, I drew this set of incubation bottle. I hope this clarify more about it. Thank you again.

Line no 113 to 115- Kindly add details about calculation of CH4 production and its CH4PP in details (Provide formula used by authors).

Thank you for your comment. The gas production potential was calculated from average cumulation of the gas emission under incubation period divided by incubation`s week as I show below.

CH4 production potential was calculated from fluctuation of gas concentration in the incubation bottle as follows:

F = r x (gas concentration x V)/(D x W) x a x 1000

where, F is the gas emission(mg.C.kg-1.day-1); r is the density of gas at the standard condition (CH4 = 0.717 kg m-3); gas concentration (ppmv); V(m3) is volume of the bottle; W(g) is dry soil weight; a = is the conversion of factor for CH4 to C (12/16). To calculate average cumulative CH4 emission, the gas emission was utilized in the formular below:

Average cumulative CH4 emission (mg C.kg-1) = F+(C x D)

where, C is the last cumulative gas result; D is the number of days in the sampling interval. 

CH4 production potential (CH4 PP) = 

Average cumulative CH4 emission (mg C.kg-1) / Number of incubation week

Table 2: remove this table from main manuscript and add to supplementary file.

Acknowledged, the table was shifted to supplementary file. Thank you

Reviewer 2 Report

Abstract: Please remove word 'sandy' from starting sentence. Sany soils may emit more, but its not a common knowledge.

Introduction: This study is all about spatial variations on emissions and finding drivers of methane production. Please add more detail about spatial variabilities in emissions and methane production potential.

Materials and method: Please add detail on how gas sampling was done from incubation chamber?

Authors comparing CH4 production potential between two sites, Fig 3, but did not present ANOVA under data analysis. Please clarify it.

Results: Authors presenting each and every results. This is redundant as they are already presented in tables/figures. please explain only important results as per objectives of the study. No need to explain every results.

Discuss: Suggest to improve discussion with particular focus on spatial variabilities--what factors are more responsible and why.

Author Response

Dear Reviewer

Thank you for the valuable comments. Please kindly find the modified manuscript attached and have a look to the reply of each comment below.

FYI : Some part of the manuscript was modified according to the other reviewers comment.

Abstract: Please remove word 'sandy' from starting sentence. Sandy soils may emit more, but its not a common knowledge.

: Thank you for your comment. The part of abstract was modified as follows:

Abstract: It is well known that submerged soils emit high levels of methane (CH4) due to oxygen deprivation and free iron oxide causing a quick reduction.

Introduction: This study is all about spatial variations on emissions and finding drivers of methane production. Please add more detail about spatial variabilities in emissions and methane production potential.

: I appreciated. I had added detail about spatial variabilities in emission and methane production potential on line no.85-94 as below:

Even though the knowledge of processes that contributes to the CH4 emission is well-reported academically, understanding on upscaling or spatial level of the emission is still inadequate [31]. Moreover, The U.S. Environmental Protection Agency (USEPA) reported in 2006 that the increasing of world population affects the demand of rice which lays an impact on methane emission especially third-fourth of the emission is emitted from south east Asian country [23]. This means the assessment of methane production potential is highly essential as it implies how soil reduction process perform [37]. With the reliable CH4 production equation that is put into potentiality and represented in spatial level, the result can imply possibility of future amount of CH4 emitted from paddy field in wide scale.

Materials and method: Please add detail on how gas sampling was done from incubation chamber?

: Thank you very much. As I do not have any clear photo, I inserted the sketch image as an example in line no.174. So this gives more understanding for readers.

Authors comparing CH4 production potential between two sites, Fig 3, but did not present ANOVA under data analysis. Please clarify it.

: Thank you. At line no.266 and 268, I added p value from one-way ANOVA analysis.

As shown in Fig 4a, there was no significant difference in the average CH4PP between Ayutthaya and Khonkaen (2012.31 and 1742.81 mg C.kg-1.week-1, respectively, p>0.05). For the soil properties measured before incubation, Ayutthaya was significantly greater than Khonkaen (p£0.05) in 1:10 ECBI (10.69 and 1.97 mS.m-1), SOM (4.26% and 1.27%), SOC (2.46% and 0.74%), ExNH4(BI) (17.07 and 12.38 mg.kg-1), ExK , ExCa, ExMg, ExNa, Total N, CEC, SO42- -SBI, and silt.

 I also attached table of one-way ANOVA result below. Would you like me to attach these tables in my manuscript?

But in my opinion, the Fig 4 and 5 are already clear with asterisk indicating the significance between groups, and the p value added at line no. 266 and 268 with confidential level of the analysis at line no. 208  had shown adequate understanding that each factor showed its significant or insignificant difference between group. Furthermore, the paper may have a redundant amount of tables and figures which overweight text. However, If this is still lacking, please kindly inform.

Table 1. Significant difference of each factors between Ayutthaya and Khonkaen using one way ANOVA (Factor analyzed after incubation)

Factors

Sum of square

df

Mean square

F

P-value

CH4PP (μgC/kg/week)

792399.57

1

792399.57

0.11

0.7

1:10 pH (b)

4.98

1

4.98

53.99

0.0

EC (1:10) (b) (mS/m-1)

829.43

1

829.43

70.16

0.0

SOM (%)

97.27

1

97.27

168.64

0.0

SOC (%)

32.47

1

32.47

168.68

0.0

Avail.P (mg kg-1)

89.18

1

89.18

0.80

0.4

Ex.K (mg kg-1)

243483.25

1

243483.25

228.95

0.0

Ex.Ca (mg kg-1)

38459884.81

1

38459884.81

75.77

0.0

Ex.Mg (mg kg-1)

2224344.40

1

2224344.40

61.86

0.0

Ex.Na (mg kg-1)

216195.70

1

216195.70

90.17

0.0

Total.N (mg kg-1)

0.10

1

0.10

73.74

0.0

CEC (cmol kg-1)

4167.80

1

4167.80

113.69

0.0

BS (%)

92.05

1

92.05

1.84

0.2

Cl- (b) (mg kg-1)

1651.23

1

1651.23

1.60

0.2

NO3- -N (b) (mg kg-1)

0.10

1

0.10

4.14

0.0

SO42- -S (b) (mg kg-1)

1621.24

1

1621.24

60.17

0.0

ExNH4 (b) (mg kg-1)

240.49

1

240.49

4.71

0.0

Sand (%)

17665.54

1

17665.54

43.52

0.0

Silt (%)

5690.43

1

5690.43

41.20

0.0

Clay (%)

3303.59

1

3303.59

39.37

0.0

Table 2. Significant difference of each factors between Ayutthaya and Khonkaen using one way ANOVA (Factor analyzed before incubation)

Factors

Sum of square

df

Mean square

F

P-value

1:10 pH (a)

0.00

1

0.00

0.08

0.8

EC (1:10) (a) (mS/m-1)

511.99

1

511.99

40.25

0.0

Fe2+ (mg kg-1)

11.68

1

11.68

0.38

0.5

Mn2+ (mg kg-1)

81.24

1

81.24

9.05

0.0

Cl- (a) (mg kg-1)

92.17

1

92.17

0.25

0.6

NO3- -N (a) (mg kg-1)

0.00

1

0.00

0.04

0.8

SO42- -S  (a) (mg kg-1)

123.62

1

123.62

8.58

0.0

ExNH4 (a)(mg kg-1)

142264.35

1

142264.35

39.79

0.0

WSOC (mg kg-1)

21.71

1

21.71

0.01

0.9

IC (mg kg-1)

101334.46

1

101334.46

10.04

0.0

Results: Authors presenting each and every results. This is redundant as they are already presented in tables/figures. please explain only important results as per objectives of the study. No need to explain every results.

: Thank you very much. I erased some of numerous information of the unnecessary factors considered only those factors that influences the significant fluctuation of gas production like CH4PP, SOM, SOC, EC, pH, ExNH4, sand content.

Discuss: Suggest to improve discussion with particular focus on spatial variabilities--what factors are more responsible and why.

: Thank you for your suggestions. Please kindly check the modification in line no. 556-568 as below:

As shown in Fig. 7 and Fig. 8, the distribution pattern of the predicted CH4PP was strongly influenced by SOC in Ayutthaya, and sand content in Khonkaen. Higher CH4PP tended to be shown at higher SOC content in Ayutthaya because it acts as electron donor which supports anaerobic reduction process. And large amount of CH4PP found in area with high sand content in Khonkaen indicating that sandy soil that contain lowest free ion among other soil can perform poorly at slowing down the reduction process [27]. Moreover, in Ayutthaya, CH4PP tended to correspond to ExNH4, which could be because ExNH4 is a product of organic matter decomposition which organic matter is an important electron donor in soil anaerobic reduction providing energy for microbial metabolism. Elevated CH4PP occurred at low 1:10 ECBI, but 1:10 ECBI in Ayutthaya fluctuated more than Khonkaen. This is because 1:10 ECBIwas influenced by SO42- reduction, one of the reduced substrates in the process indicating that lower electron contained in soil would accelerate soil reduction anaerobically. And, Ayutthaya soil contained a wide range of SO42-.
